# A new cubic transmuted power-function distribution: Properties, inference, and applications

**Muhammad Ahsan-ul-Haq**[1©], **Maha A. Aldahlan**[2©], **Javeria Zafar**[1©], **Héctor W. Gómez**[3©], **Ahmed Z. Afify**[4©]*, **Hisham A. Mahran**[5©]

**1** College of Statistical & Actuarial Sciences, University of the Punjab, Lahore, Pakistan, **2** Department of Statistics, College of Science, University of Jeddah, Jeddah, Saudi Arabia, **3** Departamento de Matemáticas, Facultad de Ciencias Básicas, Universidad de Antofagasta, Antofagasta, Chile, **4** Department of Statistics, Mathematics and Insurance, Benha University, Benha, Egypt, **5** Department of Statistics, Mathematics and Insurance, Ain Shams University, Cairo, Egypt

© These authors contributed equally to this work.

* ahmed.afify@fcom.bu.edu.eg

**Data Availability Statement:** All relevant data are within the paper.

**Funding:** The authors received no specific funding for this work.

## Abstract

A new three-parameter cubic transmuted power distribution is proposed using the cubic rank transformation. The density and hazard functions of the new distribution provide great flexibility. Some mathematical properties of the new model such as quantile function, moments, dispersion index, mean residual life, and order statistics are derived. The model parameters are estimated using five different estimation methods. A comprehensive simulation study is carried out to understand the behavior of derived estimators and choose the best estimation method. The usefulness of the proposed distribution is illustrated using a real dataset. It is concluded that the proposed distribution is better than some well-known existing distributions.

## 1. Introduction

Power-function (PF) distribution is a flexible and simple lifetime model that may offer and exhibit a better fit to some sets of failure data. The PF distribution is often employed in the assessment of semiconductor devices and electrical component reliability [1]. The PF distribution has an inverse relationship with the standard Pareto distribution and it is also a special case of Pearson type-I distribution [2]. The moments of the PF distribution are simply the negative moments of the corresponding Pareto distribution [3]. Zarrin [4] applied power function distribution to assess the component failure of semi-conductor device data by using both maximum likelihood and Bayesian estimation methods.

Meniconi and Barry [5] suggested the cumulative distribution function (cdf) and probability density function (pdf) of PF distribution are given by

$$G(x) = \left(\frac{x}{\theta}\right)^{\alpha}, \quad \alpha > 0, \theta > 0, \quad x < \theta \tag{1}$$

**Competing interests:** The authors have declared that no competing interests exist.

and

$$g(x) = \frac{\alpha x^{\alpha-1}}{\theta^{\alpha}}, \quad \alpha > 0, \theta > 0, \quad x < \theta, \tag{2}$$

where α is a shape parameter and θ is a scale parameter.

The statistical properties of the PF distribution are discussed by [2, 6, 7]. For the characterization point of view of this distribution, the authors did this task using order statistics and record values [8], using the liner function of order statistics to estimate the scale and location of the PF distribution [9]. Several authors have estimated the parameter of PF distribution. The parameter estimation of the PF distribution using moments, maximum likelihood, percentiles, least-squares methods, as well as Bayesian methods via different loss functions are discussed by [10–12]. Skaeel et al. [13] estimated the parameters of PF using L-moments, TL-moments, probability weighted moments (PWM) and generalized PWM. Bayesian analysis of PF distribution is discussed using single and double priors by [14].

Many researchers are also introduced some generalized forms of the PF distribution, for example, beta-PF [15], Kw-PF [16], Weibull-PF [17], transmuted-PF [18], transmuted Weibull PF [19], McDonald-PF [20], reflected-PF [21], Frechet-PF [22] and references therein.

In this paper, a new extended form of the PF distribution is proposed using the second-order transmuted map approach. The proposed distribution is called the new cubic transmuted power-function (NCTPF) distribution. The NCTPF distribution provides increasing, bathtub, and modified bathtub hazard rate (hr) shapes. Its density can be left-skewed, unimodal, right-skewed, concave down, concave up or reversed-J shape. Some mathematical properties are derived and studied. Five different estimation methods are used to estimate the model parameters. The flexibility of the NCTPF distribution is assessed over the PF and some of its generalizations using real-life datasets.

The paper is organized as follows. In Section 2, we introduce the new cubic transmuted power function distribution and derived some of its properties. The estimation of the NCTPF parameters using different methods is discussed in Section 3. In Section 4, a simulation study is carried out. Finally, applications on real data sets are demonstrated in Section 5 and some concluding remarks are made in Section 6.

## 2. The NCTPF distribution

A new cubic transmuted-G (NCT-G) family of distributions is proposed by [23]. The cdf of the NCT-G family is

$$F(x) = G(x)\,(1 - \lambda) + 3\,\lambda\,G^{2}(x) - 2\,\lambda\,G^{3}(x), \quad x \in R. \tag{3}$$

The corresponding pdf of the NCT-G family is obtained as

$$f(x) = g(x)\,(1 - \lambda) + 6\,\lambda\,G(x)\,g(x) - 6\,\lambda\,G^{2}(x)\,g(x), \tag{4}$$

where $\lambda \in [-1,1]$ is the transmutation parameter.

The cdf and pdf of the proposed NCTPF distribution are, respectively, given by

$$F(x) = (1 - \lambda)\left(\frac{x}{\theta}\right)^{\alpha} + 3\lambda\left(\frac{x}{\theta}\right)^{2\alpha} - 2\lambda\left(\frac{x}{\theta}\right)^{3\alpha} \tag{5}$$

and

$$f(x) = \frac{\alpha(1-\lambda)x^{\alpha-1}}{\theta^{\alpha}} + \frac{6\lambda\alpha x^{2\alpha-1}}{\theta^{2\alpha}} - \frac{6\lambda\alpha x^{3\alpha-1}}{\theta^{3\alpha}}, \quad x < \theta, \ \alpha, \ \theta > 0 \ \& \ \lambda \in [-1, 1]. \tag{6}$$

The pdf plots for various choices of the NCTPF parameters are given in Fig 1.

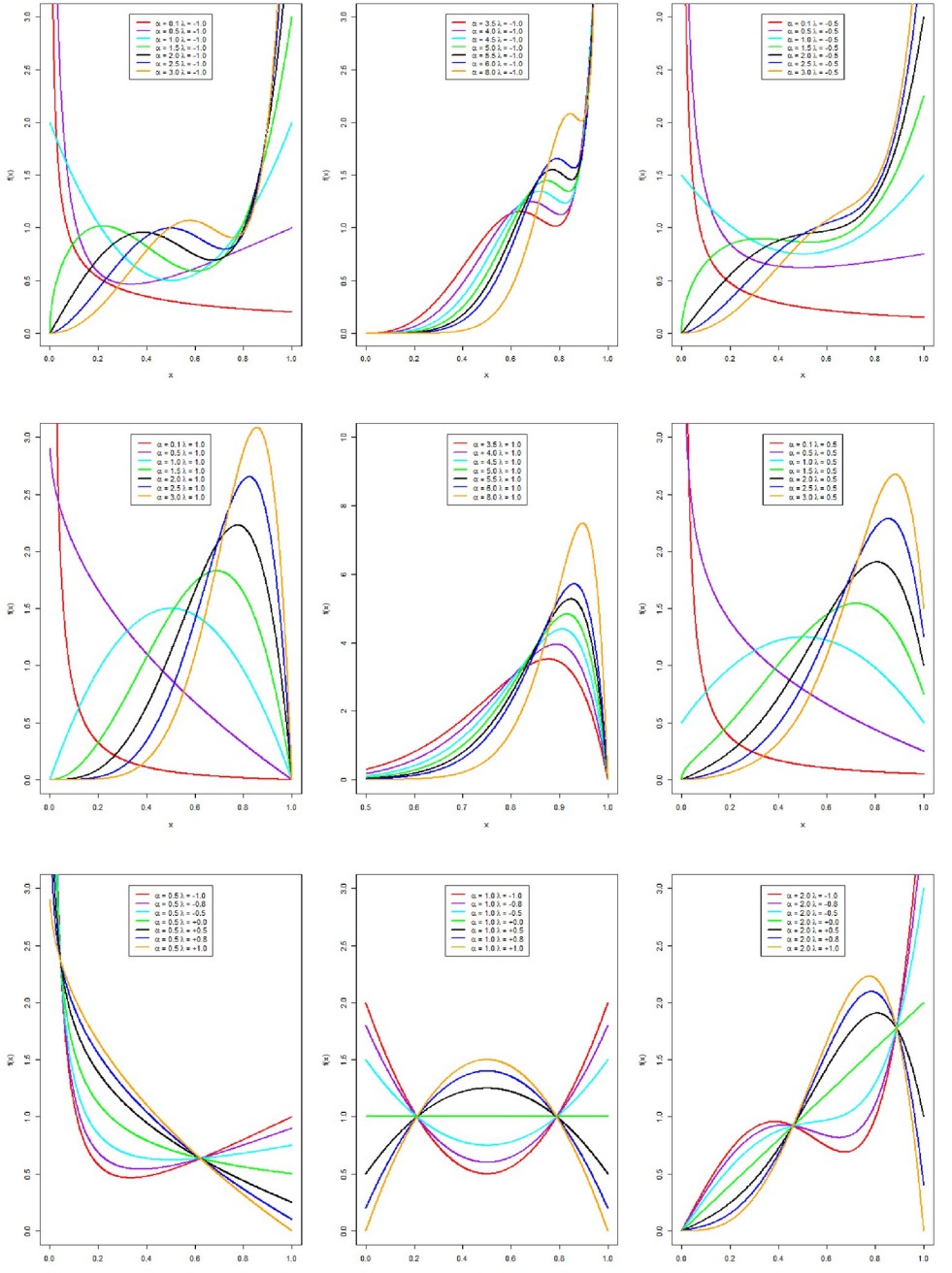

**Fig 1. The pdf curves of the NCTPF distribution for different parametric values.**

The survival function of the NCTPF distribution is

$$S(x) = 1 - \left[(1-\lambda)\left(\frac{x}{\theta}\right)^{\alpha} + 3\lambda\left(\frac{x}{\theta}\right)^{2\alpha} - 2\lambda\left(\frac{x}{\theta}\right)^{3\alpha}\right]. \tag{7}$$

The reversed hazard is defined as

$$r(x) = \frac{f(x)}{F(x)} = \frac{\alpha(1-\lambda)\theta^{2\alpha}x^{\alpha-1} + 6\lambda\alpha\theta^{\alpha}x^{2\alpha-1} - 6\lambda\alpha x^{3\alpha-1}}{(1-\lambda)\theta^{2\alpha}x^{\alpha} + 3\lambda\theta^{\alpha}x^{2\alpha} - 2\lambda x^{3\alpha}}. \tag{8}$$

The cumulative hazard function is

$$H(x) = -\log(S(x)) = -\log\left[1 - (1-\lambda)\left(\frac{x}{\theta}\right)^{\alpha} - 3\lambda\left(\frac{x}{\theta}\right)^{2\alpha} + 2\lambda\left(\frac{x}{\theta}\right)^{3\alpha}\right]. \tag{9}$$

The Mills ratio of the NCTPF distribution is

$$M = \frac{S(x)}{f(x)} = \frac{\theta^{3\alpha} - (1-\lambda)\theta^{2\alpha}x^{\alpha} - 3\lambda\theta^{\alpha}x^{2\alpha} + 2\lambda x^{3\alpha}}{\alpha(1-\lambda)\theta^{2\alpha}x^{\alpha-1} + 6\lambda\alpha\theta^{\alpha}x^{2\alpha-1} - 6\lambda\alpha x^{3\alpha-1}}. \tag{10}$$

The hr function (hrf) is

$$h(x) = \frac{\alpha(1-\lambda)\theta^{2\alpha}x^{\alpha-1} + 6\lambda\alpha\theta^{\alpha}x^{2\alpha-1} - 6\lambda\alpha x^{3\alpha-1}}{\theta^{3\alpha} - (1-\lambda)\theta^{2\alpha}x^{\alpha} - 3\lambda\theta^{\alpha}x^{2\alpha} + 2\lambda x^{3\alpha}}. \tag{11}$$

The plots of the hrf for the NCTPF distribution are given in Fig 2 which provides bathtub and increasing hrf shapes.

## 3. Mathematical properties

This section provides the derivation of some new mathematical properties of the NCTPF distribution.

### 3.1. Quantile function

The $u^{\text{th}}$ quantile of the NCTPF distribution is defined as $Q(u) = F^{-1}(u)$, where $F^{-1}(u)$ is the inverse cdf. Then, the NCTPF distribution can be simulated easily as $X = Q(U)$, where the variable $U$ has the uniform $U(0,1)$ distribution.

$$Q(u) = \frac{\theta}{2} - \frac{-6\theta^2\lambda - 3\theta^2\lambda^2}{32^{2/3}\lambda\left(54\theta^3\lambda^2 - 108u\theta^3\lambda^2 + \sqrt{4\left(-6\theta^2\lambda - 3\theta^2\lambda^2\right)^3 + \left(54\theta^3\lambda^2 - 108u\theta^3\lambda^2\right)^2}\right)^{1/3}}$$

$$+ \frac{\left(54\theta^3\lambda^2 - 108u\theta^3\lambda^2 + \sqrt{4\left(-6\theta^2\lambda - 3\theta^2\lambda^2\right)^3 + \left(54\theta^3\lambda^2 - 108u\theta^3\lambda^2\right)^2}\right)^{1/3}}{62^{1/3}\lambda}. \tag{12}$$

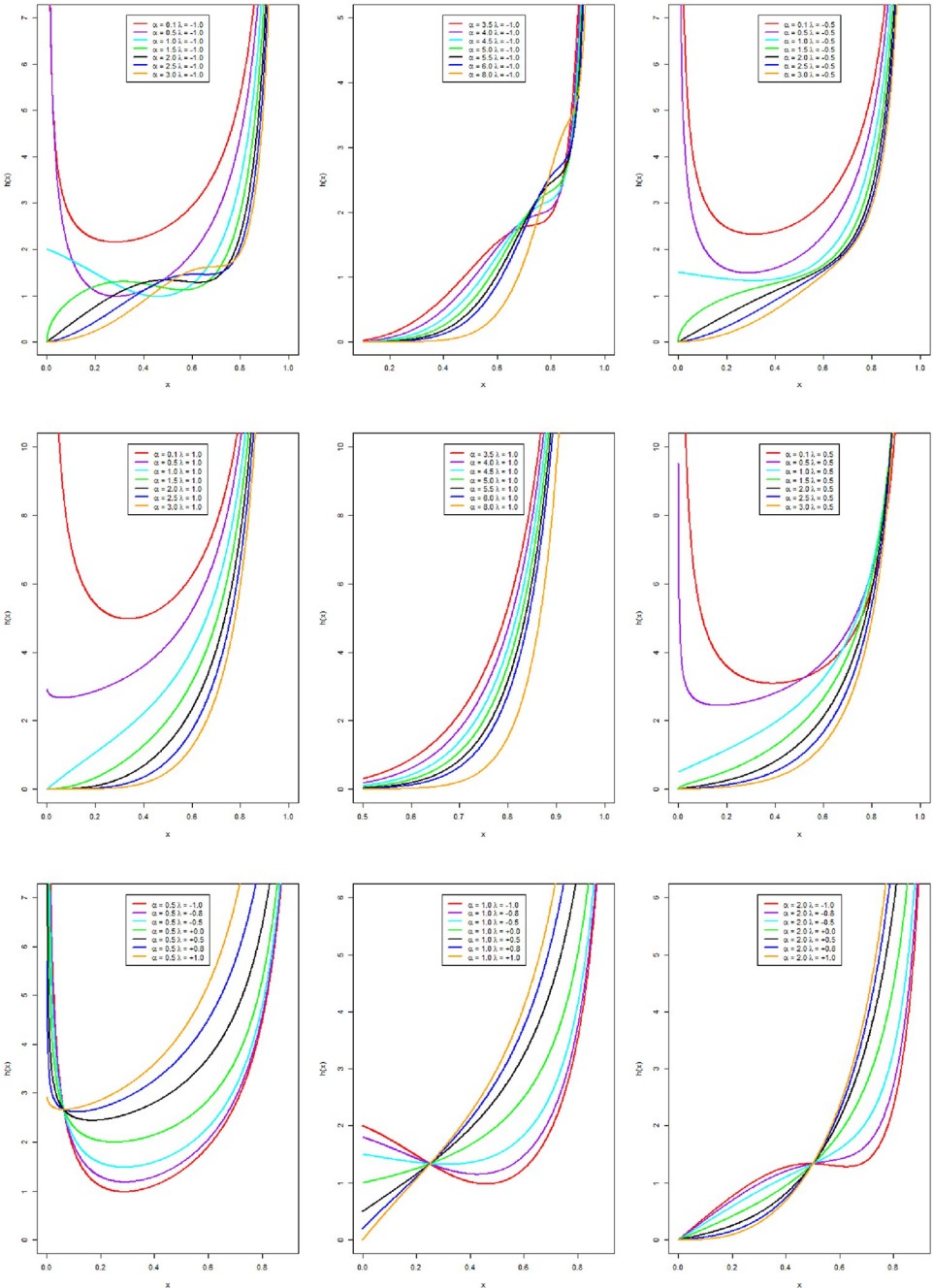

**Fig 2. The hrf curves of the NCTPF distribution for different parametric values.**

## 3.2. Moments and associated measures

A random variable X follows the NCPF distribution, then the ordinary moments can be obtained as

$$E(x^r) = \int_0^\theta x^r f(x)dx,$$

$$E(x^r) = \int_0^\theta x^r \left( \frac{\alpha(1-\lambda)x^{\alpha-1}}{\theta^\alpha} + \frac{6\lambda\alpha x^{2\alpha-1}}{\theta^{2\alpha}} - \frac{6\lambda\alpha x^{3\alpha-1}}{\theta^{3\alpha}} \right) dx.$$

After simple integration, we obtained the following final expression

$$E(x^r) = \alpha\theta^r \left[ \frac{1-\lambda}{\alpha+r} + \frac{6\lambda}{2\alpha+r} - \frac{6\lambda}{3\alpha+r} \right]. \tag{13}$$

The first four moments about the origin are

$$E(x) = \alpha\theta \left[ \frac{1-\lambda}{\alpha+1} + \frac{6\lambda}{2\alpha+1} - \frac{6\lambda}{3\alpha+1} \right], \tag{14}$$

$$E(x^2) = \alpha\theta^2 \left[ \frac{1-\lambda}{\alpha+2} + \frac{6\lambda}{2\alpha+2} - \frac{6\lambda}{3\alpha+2} \right], \tag{15}$$

$$E(x^3) = \alpha\theta^3 \left[ \frac{1-\lambda}{\alpha+3} + \frac{6\lambda}{2\alpha+3} - \frac{6\lambda}{3\alpha+3} \right] \tag{16}$$

and

$$E(x^4) = \alpha\theta^4 \left[ \frac{1-\lambda}{\alpha+4} + \frac{6\lambda}{2\alpha+4} - \frac{6\lambda}{3\alpha+4} \right]. \tag{17}$$

Based on the moments, the variance and dispersion index (DI) of X are

$$Var(X) = \alpha\theta^2 \left[ \frac{1-\lambda}{\alpha+2} + \frac{6\lambda}{2\alpha+2} - \frac{6\lambda}{3\alpha+2} \right] - \left( \alpha\theta \left[ \frac{1-\lambda}{\alpha+1} + \frac{6\lambda}{2\alpha+1} - \frac{6\lambda}{3\alpha+1} \right] \right)^2 \tag{18}$$

and

$$DI(X) = \frac{\alpha\theta^2 \left[ \frac{1-\lambda}{\alpha+2} + \frac{6\lambda}{2\alpha+2} - \frac{6\lambda}{3\alpha+2} \right] - \left( \alpha\theta \left[ \frac{1-\lambda}{\alpha+1} + \frac{6\lambda}{2\alpha+1} - \frac{6\lambda}{3\alpha+1} \right] \right)^2}{\alpha\theta \left[ \frac{1-\lambda}{\alpha+1} + \frac{6\lambda}{2\alpha+1} - \frac{6\lambda}{3\alpha+1} \right]}. \tag{19}$$

The coefficient of skewness (CS) and coefficient of kurtosis (CK) of X can be obtained using the following expressions

$$CS = \frac{E(X^3) - 3E(X)E(X^2) + 2(E(X))^3}{(Var(X))^{3/2}} \tag{20}$$

and

$$CK = \frac{E(X^4) - 4E(X)E(X^3) + 6E(X^2)(E(X))^2 - 3(E(X))^4}{(Var(X))^2}. \tag{21}$$

The mean $E(X)$, variance $Var(X)$, DI $DI(X)$, CS, and CK of the NCTPF distribution are obtained and presented in Table 1.

**Table 1. The mean, variance, DI, CS, and CK of the NCTPF distribution for ($\theta = 1$).**

| $\alpha$ | $\lambda$ | E(X) | Var(X) | DI(X) | CS | CK |
|---|---|---|---|---|---|---|
| 0.5 | -1.0 | 0.366667 | 0.122698 | 0.334632 | 0.421515 | 8.80151 |
| | -0.8 | 0.360000 | 0.116114 | 0.322540 | 0.465463 | 9.16251 |
| | -0.4 | 0.346667 | 0.102679 | 0.296190 | 0.553717 | 10.1156 |
| | +0.4 | 0.320000 | 0.074743 | 0.233571 | 0.710895 | 13.6754 |
| | +0.8 | 0.306667 | 0.060241 | 0.196439 | 0.742329 | 17.2566 |
| | +1.0 | 0.300000 | 0.052857 | 0.176190 | 0.721013 | 20.0396 |
| 1.5 | -1.0 | 0.586364 | 0.097936 | 0.167023 | -0.235456 | 75.5385 |
| | -0.8 | 0.589091 | 0.092093 | 0.156330 | -0.257062 | 86.8657 |
| | -0.4 | 0.594545 | 0.080362 | 0.135165 | -0.299904 | 117.935 |
| | +0.4 | 0.605455 | 0.056722 | 0.093684 | -0.368015 | 252.869 |
| | +0.8 | 0.610909 | 0.044812 | 0.073353 | -0.364630 | 418.597 |
| | +1.0 | 0.613636 | 0.038835 | 0.063287 | -0.334020 | 566.501 |
| 2.0 | -1.0 | 0.647619 | 0.080590 | 0.124440 | -0.399711 | 164.301 |
| | -0.8 | 0.651429 | 0.075641 | 0.116115 | -0.433841 | 190.740 |
| | -0.4 | 0.659048 | 0.065656 | 0.099623 | -0.502035 | 264.711 |
| | +0.4 | 0.674286 | 0.045339 | 0.067240 | -0.611614 | 606.063 |
| | +0.8 | 0.681905 | 0.035006 | 0.051336 | -0.600323 | 1061.54 |
| | +1.0 | 0.685714 | 0.029796 | 0.043452 | -0.538133 | 1496.95 |
| 5.0 | -1.0 | 0.814394 | 0.030040 | 0.036886 | -0.874864 | 2927.69 |
| | -0.8 | 0.818182 | 0.028058 | 0.034293 | -0.935674 | 3418.59 |
| | -0.4 | 0.825758 | 0.024007 | 0.029072 | -1.060730 | 4844.14 |
| | +0.4 | 0.840909 | 0.015561 | 0.018505 | -1.275690 | 12394.9 |
| | +0.8 | 0.848485 | 0.011166 | 0.013160 | -1.227690 | 24947.6 |
| | +1.0 | 0.852273 | 0.008925 | 0.010472 | -1.018830 | 39743.3 |

### 3.3. Mean residual life

The mean residual life is given by

$$m(t) = E(X - t|X > t) = \frac{1}{[1 - F(t)]} \int_{t}^{\infty} [1 - F(x)]dx,$$

$$m(t) = \frac{\int_{t}^{\theta} \left[ 1 - (1 - \lambda)\left(\frac{x}{\theta}\right)^{\alpha} - 3\lambda\left(\frac{x}{\theta}\right)^{2\alpha} + 2\lambda\left(\frac{x}{\theta}\right)^{3\alpha} \right] dx}{\left[ 1 - (1 - \lambda)\left(\frac{t}{\theta}\right)^{\alpha} - 3\lambda\left(\frac{t}{\theta}\right)^{2\alpha} + 2\lambda\left(\frac{t}{\theta}\right)^{3\alpha} \right]}.$$

Consider

$$\int_{t}^{\theta} \left[ 1 - (1 - \lambda)\left(\frac{x}{\theta}\right)^{\alpha} - 3\lambda\left(\frac{x}{\theta}\right)^{2\alpha} + 2\lambda\left(\frac{x}{\theta}\right)^{3\alpha} \right] dx$$

$$= \int_{t}^{\theta} 1dx - \frac{(1 - \lambda)}{\theta^{\alpha}} \int_{t}^{\theta} x^{\alpha}dx - \frac{3\lambda}{\theta^{2\alpha}} \int_{t}^{\theta} x^{2\alpha}dx + \frac{2\lambda}{\theta^{3\alpha}} \int_{t}^{\theta} x^{3\alpha}dx$$

$$= (\theta - t) - \frac{(1 - \lambda)}{\theta^{\alpha}} \left( \frac{\theta^{\alpha+1} - t^{\alpha+1}}{\alpha + 1} \right) - \frac{3\lambda}{\theta^{2\alpha}} \left( \frac{\theta^{2\alpha+1} - t^{2\alpha+1}}{2\alpha + 1} \right) + \frac{2\lambda}{\theta^{3\alpha}} \left( \frac{\theta^{3\alpha+1} - t^{3\alpha+1}}{3\alpha + 1} \right)$$

$$= \theta \left( 1 - \frac{(1 - \lambda)}{\alpha + 1} - \frac{3\lambda}{2\alpha + 1} + \frac{2\lambda}{3\alpha + 1} \right) - t \left( 1 - \frac{(1 - \lambda)t^{\alpha}}{\theta^{\alpha}(\alpha + 1)} - \frac{3\lambda t^{2\alpha}}{\theta^{2\alpha}(2\alpha + 1)} + \frac{2\lambda t^{3\alpha}}{\theta^{3\alpha}(3\alpha + 1)} \right).$$

Hence

$$m(t) = \frac{\theta\left(1 - \frac{(1-\lambda)}{\alpha+1} - \frac{3\lambda}{2\alpha+1} + \frac{2\lambda}{3\alpha+1}\right) - t\left(1 - \frac{(1-\lambda)t^\alpha}{\theta^\alpha(\alpha+1)} - \frac{3\lambda t^{2\alpha}}{\theta^{2\alpha}(2\alpha+1)} + \frac{2\lambda t^{3\alpha}}{\theta^{3\alpha}(3\alpha+1)}\right)}{\left[1 - (1-\lambda)\left(\frac{t}{\theta}\right)^\alpha - 3\lambda\left(\frac{t}{\theta}\right)^{2\alpha} + 2\lambda\left(\frac{t}{\theta}\right)^{3\alpha}\right]}. \tag{22}$$

### 3.4. Order statistics

Let $X_{i:n}$ denote the $i$th order statistic. Then, let $f_{i:n}(x)$ be the pdf of the $i$th order statistic for a random sample $X_1, X_2, \ldots, X_n$ from the NCTPF distribution. The pdf of the $i$th order statistics has the form

$$f_{i:n} = \frac{n!}{(i-1)!(n-i)!}f(x_i)[F(x_i)]^{i-1}[1-F(x_i)]^{n-i}.$$

Hence, the pdf of the $i$th order statistics of the NCTPF distribution reduces to

$$f_{i:n} = \frac{n!}{(i-1)!(n-i)!}\left[\frac{\alpha(1-\lambda)x_i^{\alpha-1}}{\theta^\alpha} + \frac{6\lambda\alpha x_i^{2\alpha-1}}{\theta^{2\alpha}} - \frac{6\lambda\alpha x_i^{3\alpha-1}}{\theta^{3\alpha}}\right]$$
$$\left[(1-\lambda)\left(\frac{x_i}{\theta}\right)^\alpha + 3\lambda\left(\frac{x_i}{\theta}\right)^{2\alpha} - 2\lambda\left(\frac{x_i}{\theta}\right)^{3\alpha}\right]^{i-1} \tag{23}$$
$$\left[1 - (1-\lambda)\left(\frac{x_i}{\theta}\right)^\alpha - 3\lambda\left(\frac{x_i}{\theta}\right)^{2\alpha} + 2\lambda\left(\frac{x_i}{\theta}\right)^{3\alpha}\right]^{n-i}.$$

From Eq (23), the minimum order statistics of the NCTPF distribution is

$$f_{1:n} = nf(x_1)[1-F(x_1)]^{n-1}$$

$$f_{1:n} = n\left[\frac{\alpha(1-\lambda)x_1^{\alpha-1}}{\theta^\alpha} + \frac{6\lambda\alpha x_1^{2\alpha-1}}{\theta^{2\alpha}} - \frac{6\lambda\alpha x_1^{3\alpha-1}}{\theta^{3\alpha}}\right]\left[1 - (1-\lambda)\left(\frac{x_1}{\theta}\right)^\alpha - 3\lambda\left(\frac{x_1}{\theta}\right)^{2\alpha} + 2\lambda\left(\frac{x_1}{\theta}\right)^{3\alpha}\right]^{n-1}.\tag{24}$$

The maximum order statistics of the NCTPF distribution reduces to

$$f_{n:n} = n\left[\frac{\alpha(1-\lambda)x_n^{\alpha-1}}{\theta^\alpha} + \frac{6\lambda\alpha x_n^{2\alpha-1}}{\theta^{2\alpha}} - \frac{6\lambda\alpha x_n^{3\alpha-1}}{\theta^{3\alpha}}\right]\left[(1-\lambda)\left(\frac{x_n}{\theta}\right)^\alpha + 3\lambda\left(\frac{x_n}{\theta}\right)^{2\alpha} - 2\lambda\left(\frac{x_n}{\theta}\right)^{3\alpha}\right]^{n-1}.\tag{25}$$

## 4. Parameter estimation of NCTPF distribution

In this section, we estimate the parameters of NCTPF distribution using six different estimation methods. The considered estimation methods are maximum likelihood (ML), weighted least-squares (WLS), ordinary least-squares (OLS), Anderson-Darling (AD), and Cramér von-Misses (CVM). A comprehensive simulation study is carried out to assess the efficient estimation method.

Let $X_1, X_2, \ldots, X_n$ be a random sample from the NCTPF distribution and $X_{(1)} < X_{(2)} < \cdots < X_{(n)}$ be the associated order statistics. Moreover, $x_{(i)}$ refers to the observed values of $X_{(i)}$.

The log-likelihood of the NCTPF distribution is

$$l(\xi) = n \ln(\alpha) + (\alpha - 1) \sum_{i=1}^{n} \ln(x_i)$$

$$+ \sum_{i=1}^{n} \ln \left[ \frac{(1-\lambda)}{\theta^{\alpha}} + \frac{6\lambda x_i^{\alpha-1}}{\theta^{2\alpha}} - \frac{6\lambda x_i^{2\alpha-1}}{\theta^{3\alpha}} \right]. \qquad (26)$$

where $\xi = (\alpha, \theta, \lambda)$ is the vector of parameters. Then, the MLE of parameters is given as follows

$$\hat{\alpha}_{MLE} = \frac{\text{argmax}(\alpha)}{\alpha}, \ \hat{\lambda}_{MLE} = \frac{\text{argmax}(\lambda)}{\lambda} \ \text{and} \ \hat{\theta}_{MLE} = \frac{\text{argmax}(\theta)}{\theta}. \qquad (27)$$

Now we define the following functions which are adopted to obtain the minimum distance-based estimates:

$$Q_{AD}(\xi) = -n - \frac{1}{n} \sum_{i=1}^{n} (2_i - 1) \left[ \log \left( (1-\lambda)\left(\frac{x_{i:n}}{\theta}\right)^{\alpha} + 3\lambda\left(\frac{x_{i:n}}{\theta}\right)^{2\alpha} - 2\lambda\left(\frac{x_{i:n}}{\theta}\right)^{3\alpha} \right) \right.$$

$$\left. + \log \left( 1 - (1-\lambda)\left(\frac{x_{i:n}}{\theta}\right)^{\alpha} + 3\lambda\left(\frac{x_{i:n}}{\theta}\right)^{2\alpha} - 2\lambda\left(\frac{x_{i:n}}{\theta}\right)^{3\alpha} \right) \right]^2$$

$$Q_{CVM}(\xi) = \frac{1}{12n} + \sum_{i=1}^{n} \left[ (1-\lambda)\left(\frac{x_{i:n}}{\theta}\right)^{\alpha} + 3\lambda\left(\frac{x_{i:n}}{\theta}\right)^{2\alpha} - 2\lambda\left(\frac{x_{i:n}}{\theta}\right)^{3\alpha} - \frac{2i-1}{2n} \right]^2,$$

$$Q_{OLS}(\xi) = \sum_{i=1}^{n} \left[ (1-\lambda)\left(\frac{x_{i:n}}{\theta}\right)^{\alpha} + 3\lambda\left(\frac{x_{i:n}}{\theta}\right)^{2\alpha} - 2\lambda\left(\frac{x_{i:n}}{\theta}\right)^{3\alpha} - \frac{i}{n+2} \right]^2$$

and

$$Q_{WLS}(\xi) = \sum_{i=1}^{n} \frac{(n+1)^2(n+2)}{i(n-i+1)} \left[ (1-\lambda)\left(\frac{x_{i:n}}{\theta}\right)^{\alpha} + 3\lambda\left(\frac{x_{i:n}}{\theta}\right)^{2\alpha} - 2\lambda\left(\frac{x_{i:n}}{\theta}\right)^{3\alpha} - \frac{i}{n+1} \right]^2.$$

The ADEs, CVMEs, OLSEs, and WLSEs of the NCTPF parameters are respectively given by

$$\hat{\alpha}_{ADE} = \frac{\text{argmax}\{Q_{AD}(\alpha)\}}{\alpha}, \ \hat{\lambda}_{ADE} = \frac{\text{argmax}\{Q_{AD}(\lambda)\}}{\lambda} \ \& \ \hat{\theta}_{ADE} = \frac{\text{argmax}\{Q_{AD}(\theta)\}}{\theta}. \qquad (28)$$

$$\hat{\alpha}_{CVME} = \frac{\text{argmax}\{Q_{CVM}(\alpha)\}}{\alpha}, \hat{\lambda}_{CVME} = \frac{\text{argmax}\{Q_{CVM}(\lambda)\}}{\lambda} \ \& \ \hat{\theta}_{CVME} = \frac{\text{argmax}\{Q_{CVM}(\theta)\}}{\theta}. \qquad (29)$$

$$\hat{\alpha}_{OLSE} = \frac{\text{argmax}\{Q_{OLS}(\alpha)\}}{\alpha}, \hat{\lambda}_{OLSE} = \frac{\text{argmax}\{Q_{OLS}(\lambda)\}}{\lambda} \ \& \ \hat{\theta}_{OLSE} = \frac{\text{argmax}\{Q_{OLS}(\theta)\}}{\theta}. \quad (30)$$

$$\hat{\alpha}_{WSE} = \frac{\text{argmax}\{Q_{WLS}(\alpha)\}}{\alpha}, \hat{\lambda}_{WLSE} = \frac{\text{argmax}\{Q_{WLS}(\lambda)\}}{\lambda} \ \& \ \hat{\theta}_{WLSE} = \frac{\text{argmax}\{Q_{WLS}(\theta)\}}{\theta}. \quad (31)$$

The estimators presented in Eqs (28)-(31) can be obtained by using the **optim ()** function in R.

## 5. Simulation

This section provides a comprehensive simulation study to explore and compare the performance of the suggested estimators. The samples are generated from NCTPF distribution with sample sizes $n$ = 20, 50, 100, 200, 300, and some combination of parameters $\lambda$ and $\alpha$ with $\theta$ = 1. The simulation procedure is based on 10,000 repetitions. The performance of these estimators is assessed using absolute bias (AB) and mean square errors (MSE) which are calculated using the R software. The AB and MSE are presented in Tables 2–6.

The results illustarte that the average estimates are closer to the true values of the parameters as the sample size increases. Further, the ABs and MSEs for all estimates decrease with the increase in sample size. The five methods demonstrate the consistency property. We conclude that the ML approach performs well in predicting the parameters of the NCTPF distribution.

**Table 2. Simulation results for ($\lambda$ = -0.9,$\alpha$ = 0.5).**

| $n$ | Par. | Est. | MLE | ADE | CVME | OLSE | WLSE |
|---|---|---|---|---|---|---|---|
| 20 | $\hat{\lambda}$ | AB | 0.19410 | 0.23878 | 0.29807 | 0.19645 | 0.20620 |
| | | MSE | 0.21742 | 0.23260 | 0.30804 | 0.20546 | 0.21579 |
| | $\hat{\alpha}$ | AB | 0.00918 | 0.00493 | 0.02228 | 0.01711 | 0.01131 |
| | | MSE | 0.01534 | 0.02162 | 0.03607 | 0.03417 | 0.02824 |
| 50 | $\hat{\lambda}$ | AB | 0.08316 | 0.11765 | 0.13678 | 0.09767 | 0.10669 |
| | | MSE | 0.06793 | 0.08622 | 0.09905 | 0.07765 | 0.08357 |
| | $\hat{\alpha}$ | AB | 0.00335 | 0.00210 | 0.00864 | 0.00669 | 0.00395 |
| | | MSE | 0.00514 | 0.00799 | 0.01129 | 0.01095 | 0.00863 |
| 100 | $\hat{\lambda}$ | AB | 0.06367 | 0.08722 | 0.09346 | 0.07242 | 0.08244 |
| | | MSE | 0.03685 | 0.04914 | 0.05124 | 0.04365 | 0.04822 |
| | $\hat{\alpha}$ | AB | 0.00012 | 0.00166 | 0.00102 | 0.00016 | 0.00072 |
| | | MSE | 0.00255 | 0.00401 | 0.00527 | 0.00519 | 0.00415 |
| 200 | $\hat{\lambda}$ | AB | 0.01992 | 0.03395 | 0.03709 | 0.02664 | 0.03160 |
| | | MSE | 0.01735 | 0.02135 | 0.02147 | 0.01948 | 0.02118 |
| | $\hat{\alpha}$ | AB | 0.00010 | 0.00011 | 0.00127 | 0.00084 | 0.00047 |
| | | MSE | 0.00124 | 0.00189 | 0.00246 | 0.00244 | 0.00192 |
| 300 | $\hat{\lambda}$ | AB | 0.02400 | 0.03227 | 0.03324 | 0.02555 | 0.03044 |
| | | MSE | 0.01351 | 0.01672 | 0.01610 | 0.01500 | 0.01650 |
| | $\hat{\alpha}$ | AB | 0.00053 | 0.00035 | 0.00091 | 0.00064 | 0.00000 |
| | | MSE | 0.00072 | 0.00117 | 0.00154 | 0.00153 | 0.00119 |

**Table 3. Simulation results for ($\lambda$ = -0.5,$\alpha$ = 0.5).**

| n | Par. | Est. | MLE | ADE | CVME | OLSE | WLSE |
|---|---|---|---|---|---|---|---|
| 20 | $\hat{\lambda}$ | AB | 0.07198 | 0.07742 | 0.15269 | 0.01728 | 0.03318 |
| | | MSE | 0.29952 | 0.25602 | 0.30297 | 0.23557 | 0.24637 |
| | $\hat{\alpha}$ | AB | 0.02185 | 0.01914 | 0.03400 | 0.03006 | 0.02512 |
| | | MSE | 0.01955 | 0.02271 | 0.03289 | 0.03124 | 0.02703 |
| 50 | $\hat{\lambda}$ | AB | 0.02664 | 0.03897 | 0.06816 | 0.00727 | 0.02195 |
| | | MSE | 0.12452 | 0.12155 | 0.12542 | 0.11337 | 0.12107 |
| | $\hat{\alpha}$ | AB | 0.00566 | 0.00251 | 0.00653 | 0.00556 | 0.00371 |
| | | MSE | 0.00603 | 0.00763 | 0.00955 | 0.00931 | 0.00790 |
| 100 | $\hat{\lambda}$ | AB | 0.00163 | 0.00495 | 0.02035 | 0.01123 | 0.00195 |
| | | MSE | 0.06214 | 0.06649 | 0.06512 | 0.06331 | 0.06673 |
| | $\hat{\alpha}$ | AB | 0.00559 | 0.00503 | 0.00707 | 0.00668 | 0.00588 |
| | | MSE | 0.00309 | 0.00421 | 0.00516 | 0.00510 | 0.00437 |
| 200 | $\hat{\lambda}$ | AB | 0.00162 | 0.00486 | 0.01203 | 0.00404 | 0.00152 |
| | | MSE | 0.02807 | 0.03091 | 0.02897 | 0.02865 | 0.03116 |
| | $\hat{\alpha}$ | AB | 0.00449 | 0.00394 | 0.00489 | 0.00474 | 0.00437 |
| | | MSE | 0.00149 | 0.00193 | 0.00232 | 0.00231 | 0.00196 |
| 300 | $\hat{\lambda}$ | AB | 0.00049 | 0.00223 | 0.00894 | 0.00179 | 0.00006 |
| | | MSE | 0.01914 | 0.02131 | 0.02037 | 0.02021 | 0.02145 |
| | $\hat{\alpha}$ | AB | 0.00321 | 0.00221 | 0.00249 | 0.00239 | 0.00254 |
| | | MSE | 0.00100 | 0.00131 | 0.00157 | 0.00156 | 0.00133 |

**Table 4. Simulation results for ($\lambda$ = 0.5,$\alpha$ = 0.5).**

| n | Par. | Est. | MLE | ADE | CVME | OLSE | WLSE |
|---|---|---|---|---|---|---|---|
| 20 | $\hat{\lambda}$ | AB | 0.04699 | 0.02265 | 0.03967 | 0.10854 | 0.08153 |
| | | MSE | 0.20540 | 0.16836 | 0.18057 | 0.21585 | 0.19669 |
| | $\hat{\alpha}$ | AB | 0.01779 | 0.01342 | 0.01483 | 0.01414 | 0.01394 |
| | | MSE | 0.01545 | 0.01436 | 0.01519 | 0.01499 | 0.01460 |
| 50 | $\hat{\lambda}$ | AB | 0.01037 | 0.02071 | 0.01248 | 0.05557 | 0.03284 |
| | | MSE | 0.08701 | 0.08098 | 0.09537 | 0.10254 | 0.08782 |
| | $\hat{\alpha}$ | AB | 0.00350 | 0.00276 | 0.00336 | 0.00321 | 0.00267 |
| | | MSE | 0.00435 | 0.00427 | 0.00454 | 0.00452 | 0.00432 |
| 100 | $\hat{\lambda}$ | AB | 0.00928 | 0.00727 | 0.01360 | 0.02313 | 0.00971 |
| | | MSE | 0.04669 | 0.04422 | 0.05360 | 0.05395 | 0.04693 |
| | $\hat{\alpha}$ | AB | 0.00635 | 0.00625 | 0.00648 | 0.00645 | 0.00629 |
| | | MSE | 0.00245 | 0.00239 | 0.00248 | 0.00247 | 0.00241 |
| 200 | $\hat{\lambda}$ | AB | 0.01116 | 0.00336 | 0.01512 | 0.00357 | 0.00373 |
| | | MSE | 0.02224 | 0.02241 | 0.02752 | 0.02719 | 0.02316 |
| | $\hat{\alpha}$ | AB | 0.00022 | 0.00029 | 0.00020 | 0.00020 | 0.00033 |
| | | MSE | 0.00116 | 0.00116 | 0.00121 | 0.00121 | 0.00117 |
| 300 | $\hat{\lambda}$ | AB | 0.00365 | 0.00034 | 0.00755 | 0.00495 | 0.00089 |
| | | MSE | 0.01423 | 0.01484 | 0.01839 | 0.01829 | 0.01523 |
| | $\hat{\alpha}$ | AB | 0.00117 | 0.00110 | 0.00116 | 0.00116 | 0.00104 |
| | | MSE | 0.00073 | 0.00075 | 0.00079 | 0.00079 | 0.00075 |

**Table 5. Simulation results for ($\lambda = 0.5, \alpha = 2.0$).**

| n | Par. | Est. | MLE | ADE | CVME | OLSE | WLSE |
|---|---|---|---|---|---|---|---|
| 20 | $\hat{\lambda}$ | AB | 0.01841 | 0.06838 | 0.01057 | 0.13906 | 0.11620 |
| | | MSE | 0.23341 | 0.19083 | 0.20118 | 0.24423 | 0.22361 |
| | $\hat{\alpha}$ | AB | 0.08901 | 0.06406 | 0.07021 | 0.06751 | 0.06507 |
| | | MSE | 0.23296 | 0.20448 | 0.22025 | 0.21790 | 0.20989 |
| 50 | $\hat{\lambda}$ | AB | 0.02999 | 0.00620 | 0.02780 | 0.04145 | 0.01943 |
| | | MSE | 0.09275 | 0.08528 | 0.10182 | 0.10563 | 0.09221 |
| | $\hat{\alpha}$ | AB | 0.01286 | 0.01466 | 0.01910 | 0.01850 | 0.01536 |
| | | MSE | 0.07148 | 0.07232 | 0.07669 | 0.07635 | 0.07305 |
| 100 | $\hat{\lambda}$ | AB | 0.00105 | 0.01665 | 0.00423 | 0.03211 | 0.01834 |
| | | MSE | 0.04335 | 0.04427 | 0.05451 | 0.05592 | 0.04661 |
| | $\hat{\alpha}$ | AB | 0.00472 | 0.00326 | 0.00348 | 0.00339 | 0.00338 |
| | | MSE | 0.03523 | 0.03554 | 0.03720 | 0.03714 | 0.03572 |
| 200 | $\hat{\lambda}$ | AB | 0.00528 | 0.00397 | 0.00603 | 0.01269 | 0.00316 |
| | | MSE | 0.02257 | 0.02300 | 0.02814 | 0.02810 | 0.02343 |
| | $\hat{\alpha}$ | AB | 0.00385 | 0.00290 | 0.00235 | 0.00235 | 0.00289 |
| | | MSE | 0.01760 | 0.01806 | 0.01902 | 0.01900 | 0.01810 |
| 300 | $\hat{\lambda}$ | AB | 0.00036 | 0.00393 | 0.00459 | 0.00790 | 0.00313 |
| | | MSE | 0.01411 | 0.01536 | 0.01943 | 0.01939 | 0.01567 |
| | $\hat{\alpha}$ | AB | 0.00514 | 0.00494 | 0.00515 | 0.00514 | 0.00501 |
| | | MSE | 0.01209 | 0.01228 | 0.01284 | 0.01283 | 0.01231 |

**Table 6. Simulation results for ($\lambda = 0.5, \alpha = 3.0$).**

| n | Par. | Est. | MLE | ADE | CVME | OLSE | WLSE |
|---|---|---|---|---|---|---|---|
| 20 | $\hat{\lambda}$ | AB | 0.04157 | 0.01725 | 0.05137 | 0.09668 | 0.07195 |
| | | MSE | 0.20325 | 0.16777 | 0.18085 | 0.21307 | 0.19612 |
| | $\hat{\alpha}$ | AB | 0.10903 | 0.07717 | 0.09080 | 0.08598 | 0.08374 |
| | | MSE | 0.49833 | 0.46742 | 0.52148 | 0.51175 | 0.48439 |
| 50 | $\hat{\lambda}$ | AB | 0.01000 | 0.01901 | 0.01687 | 0.05248 | 0.03094 |
| | | MSE | 0.08704 | 0.07851 | 0.09063 | 0.09611 | 0.08436 |
| | $\hat{\alpha}$ | AB | 0.03946 | 0.03032 | 0.03234 | 0.03173 | 0.03073 |
| | | MSE | 0.18188 | 0.17175 | 0.17750 | 0.17695 | 0.17240 |
| 100 | $\hat{\lambda}$ | AB | 0.01039 | 0.00518 | 0.01605 | 0.02058 | 0.00916 |
| | | MSE | 0.04497 | 0.04417 | 0.05596 | 0.05621 | 0.04660 |
| | $\hat{\alpha}$ | AB | 0.01541 | 0.01341 | 0.01353 | 0.01342 | 0.01335 |
| | | MSE | 0.08369 | 0.08289 | 0.08634 | 0.08619 | 0.08313 |
| 200 | $\hat{\lambda}$ | AB | 0.00507 | 0.01329 | 0.00316 | 0.02184 | 0.01380 |
| | | MSE | 0.02165 | 0.02250 | 0.02751 | 0.02782 | 0.02297 |
| | $\hat{\alpha}$ | AB | 0.00609 | 0.00664 | 0.00778 | 0.00775 | 0.00691 |
| | | MSE | 0.03847 | 0.03988 | 0.04248 | 0.04242 | 0.03998 |
| 300 | $\hat{\lambda}$ | AB | 0.00788 | 0.00247 | 0.00921 | 0.00328 | 0.00291 |
| | | MSE | 0.01357 | 0.01395 | 0.01708 | 0.01694 | 0.01416 |
| | $\hat{\alpha}$ | AB | 0.00421 | 0.00463 | 0.00496 | 0.00496 | 0.00468 |
| | | MSE | 0.02858 | 0.02881 | 0.02986 | 0.02984 | 0.02886 |

**Table 7. Descriptive statistics for the data sets.**

| n | Minimum | Median | Mean | Maximum | Variance | Skewness | Kurtosis |
|---|---------|--------|------|---------|----------|----------|----------|
| **30** | 0.0200 | 0.5065 | 0.4940 | 0.9900 | 0.1417 | 0.0649 | 1.3129 |

## 6. Application to lifetime data

In this section, we analyze lifetime data to illustrate the flexibility and usefulness of the proposed distribution using the R software. For this, we consider a lifetime dataset.

The dataset is about the lifetimes of 30 electronic devices. The data observations are: 0.020, 0.029, 0.034, 0.044, 0.057, 0.096, 0.106, 0.139, 0.156, 0.164, 0.167, 0.177, 0.250, 0.326, 0.406, 0.607, 0.650, 0.672, 0.676, 0.736, 0.817, 0.838, 0.910, 0.931, 0.946, 0.953, 0.961, 0.981, 0.982 and 0.990.

Some descriptive statistics of these datasets are presented in Table 7. The boxplots and TTT plots are given in Fig 3. The TTT plot shows that the data has a bathtub hr shape. Hence, the NCTPF distribution is a suitable model for fitting this real-life data because its hrf provides bathtub hrf shape.

The NCTPF distribution is compared with some renowned competitive distributions, namely, cubic transmuted PF (CTPF), Kumaraswamy (Kw), beta, and PF distributions.

The model parameters of the NCTPFD and all competitor distributions are estimated using the maximum likelihood approach. The criterions log-likelihood, Akaike information criterion (AIC), Bayesian information criterion (BIC), along with the goodness-of-fit statistics such as Anderson-Darling (AD), Cramer von-Misses (CVM), and Kolmogorov-Smirnov (KS) with respective p-value are used. The parameter estimates and goodness-of-fit measures are given in Table 8.

The fitted probability density function, cumulative distribution function, survival function, and P-P plots for the NCTPF distribution for the lifetime of electronics devices dataset are represented in Fig 4. The findings in Table 8 and Fig 4 show that the NCTPF distribution is the

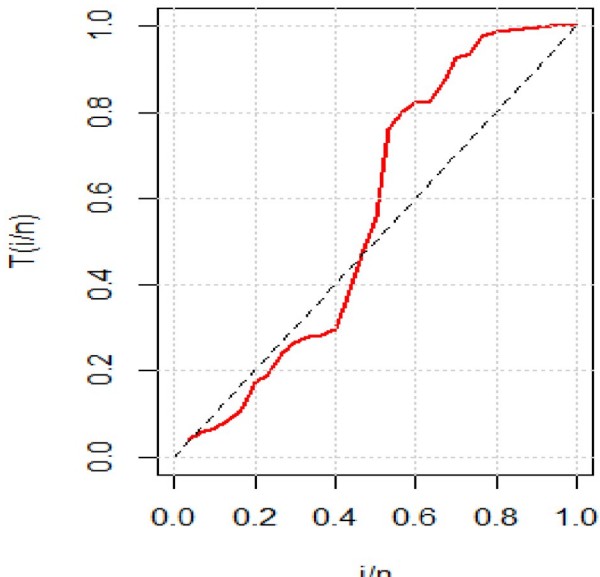
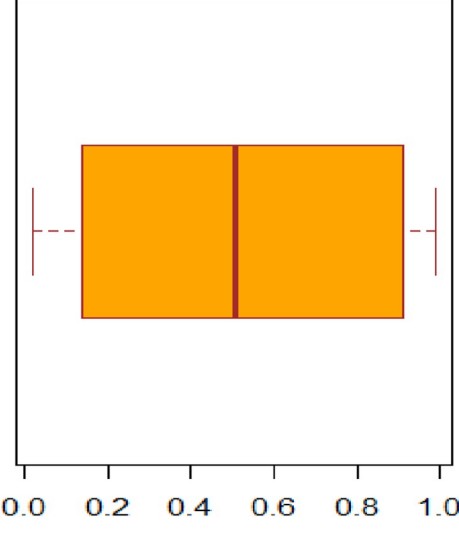

**Fig 3. Boxplot and TTT plots for both datasets.**

**Table 8. ML estimates, AIC, BIC statistics under-considered distributions based on data set.**

| Model | MLES | LogL | AIC | BIC | KS (p-value) |
|---|---|---|---|---|---|
| NCTPF | $\hat{\alpha} = 0.9715$ | 6.2151 | -8.4301 | -5.6277 | 0.1189 (0.7455) |
| | $\hat{\lambda} = -1.000$ | | | | |
| CTPF | $\hat{\alpha} = 0.9919$ | 2.6769 | -1.3538 | 1.4486 | 9.2265 (0.0018) |
| | $\hat{\lambda} = -1.000$ | | | | |
| Kw | $\hat{\alpha} = 0.8963$ | 3.5025 | -3.0050 | -0.2026 | 0.1600 (0.3850) |
| | $\hat{\lambda} = 0.3369$ | | | | |
| Beta | $\hat{\alpha} = 0.8963$ | 3.6248 | -3.2498 | -0.4474 | 0.1550 (0.4051) |
| | $\hat{\lambda} = 0.3369$ | | | | |
| PF | $\hat{\alpha} = 0.8165$ | 0.6599 | 0.6801 | 2.0813 | 3.2467 (0.1116) |

best fit compared to other distributions that are comparable. The results show that the CRTPF distribution provides a significantly better fit as compared to other models.

Another aim of this paper is to identify the selection of the best estimation technique. We also estimate the parameters of the NCTPF distribution using different estimation methods which are discussed in Section 4. Table 9 presents the parameter estimates and Kolmogorov-Smirnov statistics along with p-values for all estimation methods. Fig 4 represents P-P plots for

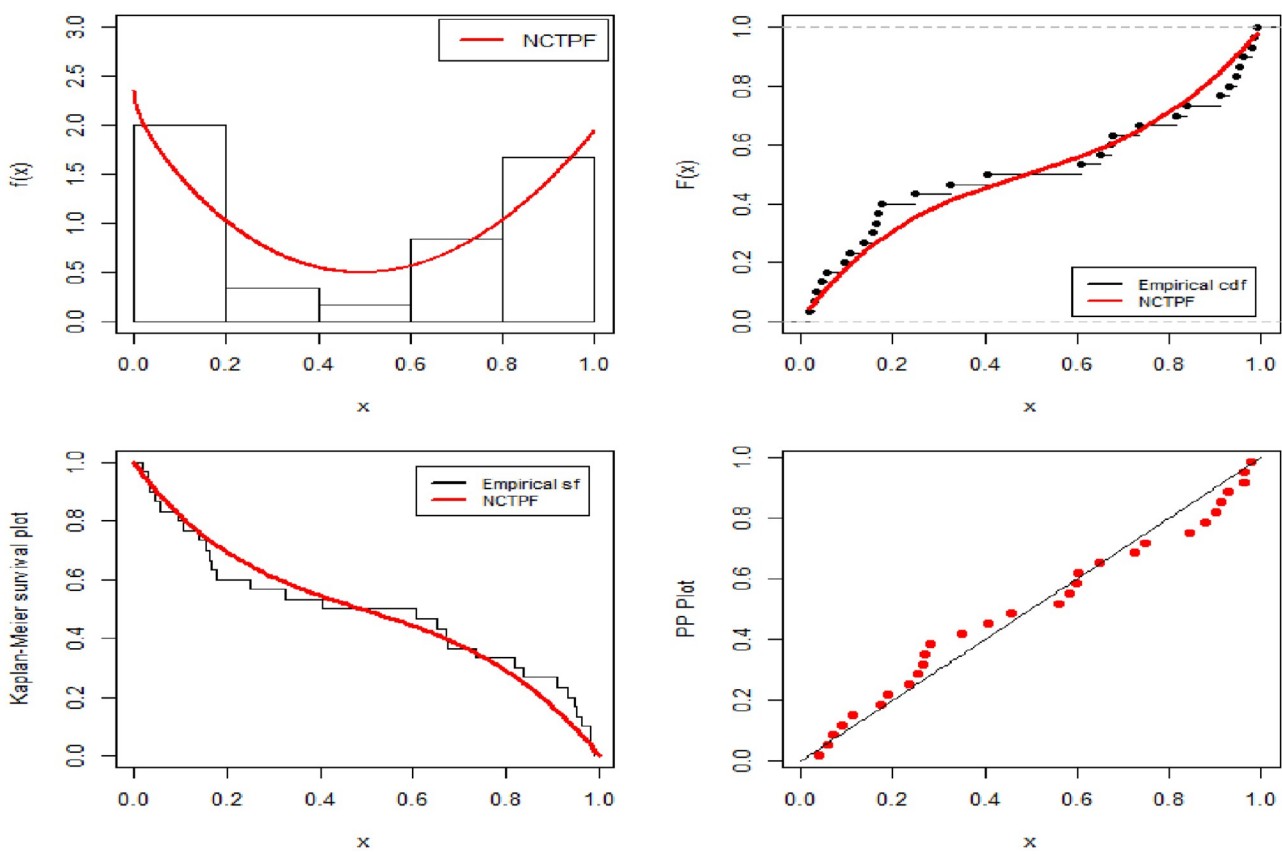

**Fig 4. Fitting performance of the NCTPF distribution.**

**Table 9. Different estimation methods for the dataset.**

| Method ↓ Statistics → | $\alpha$ | $\lambda$ | KS | P-value |
|---|---|---|---|---|
| ADE | 0.95359 | -1.0000 | 0.11557 | 0.7756 |
| CVME | 0.92097 | -1.0000 | 0.11989 | 0.7372 |
| OLSE | 0.90715 | -1.0000 | 0.12174 | 0.7204 |
| WLSE | 0.92484 | -1.0000 | 0.11938 | 0.7418 |

the NCTPF distribution by using different estimation methods. Fig 5 shows the probability-probability (PP) plots for the dataset using the different estimates in Table 9.

## 7. Conclusion

In this paper, a three-parameter distribution called the new cubic transmuted-power function (NCTPF) distribution has been introduced. The main feature of the two-parameters NCTPF distribution is its ability to model real-life data with bathtub hazard rate. A detailed study on

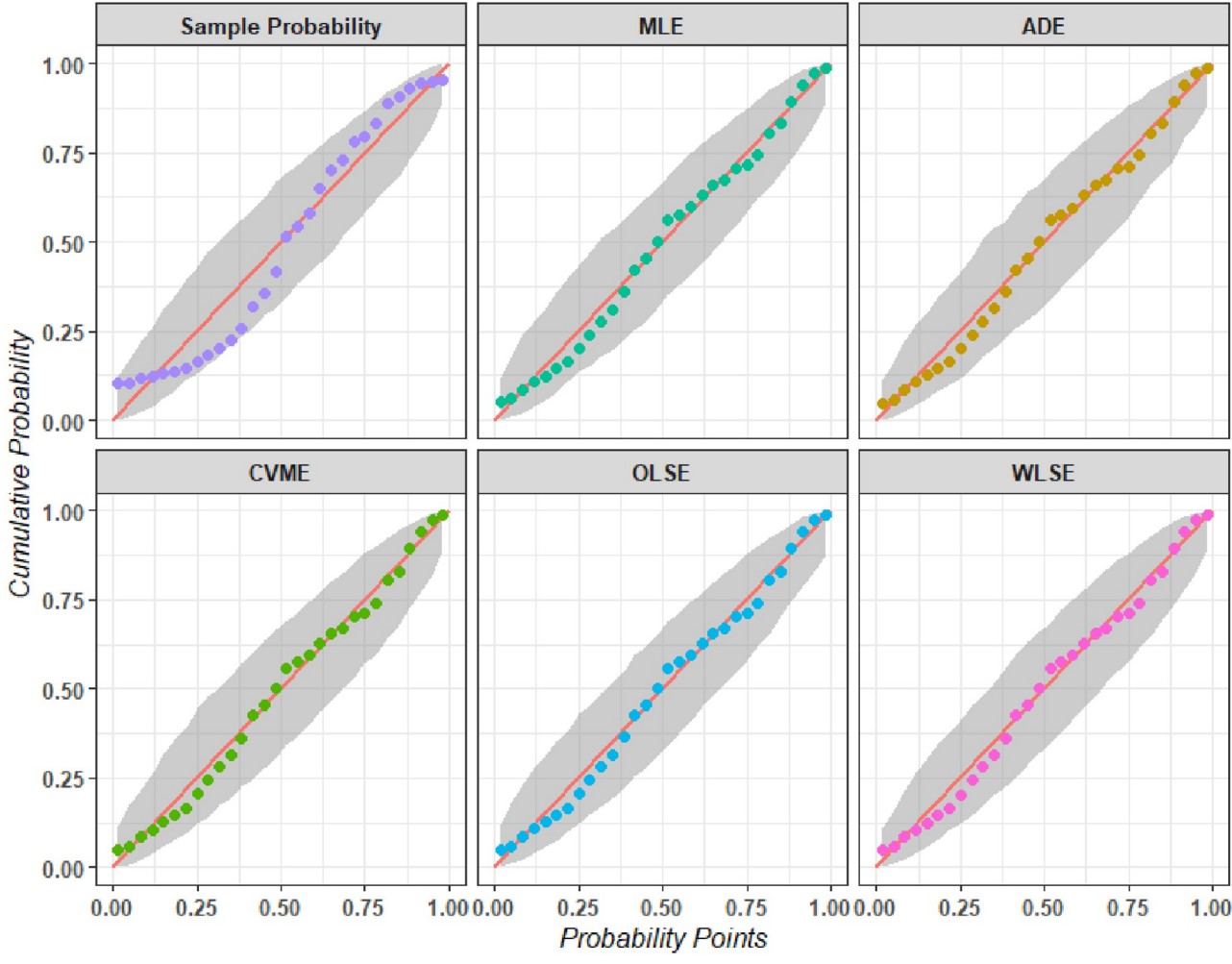

**Fig 5. The PP plots for the dataset using different estimation methods.**

the mathematical properties of the NCTPF distribution has been presented. We derived the survival and hazard functions, order statistics, quantile function, skewness, and kurtosis. The NCTPF parameters are estimated using five different estimation methods. The results show that the maximum likelihood is recommended to estimate the NCTPF parameters. A real-life data application shows that the NCTPF can be adopted effectively to provide better fits than other competing distributions. We hope that the NCTPF model may attract wider applications in applied areas to model real-life data with bathtub and modified bathtub shapes.

## Acknowledgments

The authors would like to thank the editorial board, and two reviewers for their constructive comments which greatly improved the final version of this paper.

## Author Contributions

**Conceptualization:** Maha A. Aldahlan, Héctor W. Gómez, Ahmed Z. Afify.

**Formal analysis:** Hisham A. Mahran.

**Investigation:** Maha A. Aldahlan, Javeria Zafar, Héctor W. Gómez, Hisham A. Mahran.

**Methodology:** Maha A. Aldahlan, Javeria Zafar, Héctor W. Gómez, Ahmed Z. Afify, Hisham A. Mahran.

**Project administration:** Ahmed Z. Afify.

**Resources:** Muhammad Ahsan-ul-Haq, Héctor W. Gómez.

**Software:** Muhammad Ahsan-ul-Haq, Javeria Zafar.

**Validation:** Hisham A. Mahran.

**Visualization:** Javeria Zafar.

**Writing – original draft:** Muhammad Ahsan-ul-Haq, Javeria Zafar, Hisham A. Mahran.

**Writing – review & editing:** Maha A. Aldahlan, Héctor W. Gómez, Ahmed Z. Afify.

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
