## [Decision Letter · Decision Letter 0]

19 Dec 2022

PONE-D-22-31175A New Cubic Transmuted Power-Function Distribution: Properties, Inference, and ApplicationsPLOS ONE

Dear Dr. Afify,

Thank you for submitting your manuscript to PLOS ONE. After careful consideration, we feel that it has merit but does not fully meet PLOS ONE’s publication criteria as it currently stands. Therefore, we invite you to submit a revised version of the manuscript that addresses the points raised during the review process.

ACADEMIC EDITOR: Please following reviewers' comments and proof read the revised version before resubmission

We look forward to receiving your revised manuscript.

Kind regards,

Qichun Zhang, PhD

Academic Editor

PLOS ONE

Journal Requirements:

3. Please ensure that you refer to Figure 5 in your text as, if accepted, production will need this reference to link the reader to the figure.

Additional Editor Comments:

The paper is well written with publishable contents. However, some points have been noticed by reviewers, where the novelty of the paper needs to be further highlighted. The motivation is not clear and the impact of parameter selection should be discussed and analysed carefully.

Reviewers' comments:

Reviewer's Responses to Questions

**Comments to the Author**

1. Is the manuscript technically sound, and do the data support the conclusions?

Reviewer #1: Yes

Reviewer #2: Yes

2. Has the statistical analysis been performed appropriately and rigorously? 

Reviewer #1: Yes

Reviewer #2: Yes

3. Have the authors made all data underlying the findings in their manuscript fully available?

Reviewer #1: Yes

Reviewer #2: Yes

4. Is the manuscript presented in an intelligible fashion and written in standard English?

Reviewer #1: Yes

Reviewer #2: Yes

5. Review Comments to the Author

Reviewer #1: The authors, in this paper, proposed a new three-parameter cubic transmuted power distribution for which some

mathematical properties are derived. I have some major concerns about this paper. Please rearrange the paper using following comments.

1. The numerical technique is not properly discussed. Which numerical method or technique is used to attain the numerical solution of mathematical model.

2. Abstract does not describe the major theme of the paper.

3. You have demonstrated real data sets. Please elaborate their impacts on parameters.

4. How can be the performance of proposed estimators measured.

5. What are the main features of new cubic transmuted-power function. Discuss in Conclusion section.

6. What is new in the proposed mathematical model and why it is considered.

Reviewer #2: Overall, the manuscript is technically sound. However, the writing in the abstract, introduction and conclusion needs some work in order to reach an acceptable level for publication in this journal. Some examples:

1. In the abstract, the authors mention that some mathematical properties are derived. The properties should be explicitly mentioned for many reasons.

2.Page 2:Middle paragraph starts with "Researchers mention..." This is not good writing. The authors should completely rewrite this paragraph.

3. In the conclusion, the first paragraph starts with "In this study..." Throughout the manuscript, the authors refer to it as a paper, study etc. These are informal terms and need some polishing.

6. PLOS authors have the option to publish the peer review history of their article (what does this mean?). If published, this will include your full peer review and any attached files.

Reviewer #1: **Yes: **Sohail Ahmad

Reviewer #2: No

---

## [Author Response · Author response to Decision Letter 0]

5 Jan 2023

Dear Professor Editor,

Enclosed herewith are the pdf of the revised version of our paper entitled “A New Cubic Transmuted Power-Function Distribution: Properties, Inference, and Applications”, which we hope you will find now satisfactory for publication in PloS One.

First, we would like to thank the Editor and the two reviewers for very constructive comments. In the revised version all suggestions and comments have been taken into account and addressed. All corrections and modifications are incorporated in the revised version and highlighted in RED color. 

We now answer the comments made by the editor and reviewers in the order they appeared in the reports.

Additional requirements:

Answer: We did our best to ensure that the paper is prepared according to PloS ONE style.

Answer: This is not applicable in our case. The data set is already mentioned in the manuscript. 

3. Please ensure that you refer to Figure 5 in your text as, if accepted, production will need this reference to link the reader to the figure.

Answer: Thank you. This typo is corrected.

Additional Editor Comments:

The paper is well written with publishable contents. However, some points have been noticed by reviewers, where the novelty of the paper needs to be further highlighted. The motivation is not clear and the impact of parameter selection should be discussed and analysed carefully.

Answer: Many thanks for these comments. We have addressed each and every comment raised by the two reviewers. All corrections have been incorporated in RED color.

Reviewers’ Comments to the Authors:

Reviewer #1: The authors, in this paper, proposed a new three-parameter cubic transmuted power distribution for which some mathematical properties are derived. I have some major concerns about this paper. Please rearrange the paper using following comments.

1. The numerical technique is not properly discussed. Which numerical method or technique is used to attain the numerical solution of mathematical model.

Answer: Thanks for this comment. The R software is used to obtain simulation results and the empirical results. We mentioned this in Sections 5 and 6.

2. Abstract does not describe the major theme of the paper.

Answer: Thank you. The abstract is improved.

3. You have demonstrated real data sets. Please elaborate their impacts on parameters.

Answer: The estimates of the parameters in the application show a good fit to the electronic devices data, as shown in Figure 4.

4. How can be the performance of proposed estimators measured.

Answer: Thanks for this comment. The performance of the proposed estimators can be explored and measured using simulation results through calculating some measures such as the average biases (AB) and mean square errors (MSE) for all studied cases as shown in Tables 2-6.

5. What are the main features of new cubic transmuted-power function. Discuss in conclusion section.

Answer: Thanks for this comment. It is mentioned in the conclusion section.

6. What is new in the proposed mathematical model and why it is considered.

Answer: The idea behind this construction is simple and it aims to obtain more flexible models that adapt to empirical data distributions. It is not easy to find pdf and cdf that are shaped like a bathtub and the NCTPF model has that property.

Reviewer #2: Overall, the manuscript is technically sound. However, the writing in the abstract, introduction and conclusion needs some work in order to reach an acceptable level for publication in this journal. Some examples:

1. In the abstract, the authors mention that some mathematical properties are derived. The properties should be explicitly mentioned for many reasons.

Answer: Thanks for your comment. We have added them in the abstract.

2. Page 2: Middle paragraph starts with "Researchers mention..." This is not good writing. The authors should completely rewrite this paragraph.

Answer: Thanks for careful reading. We have corrected them.

3. In the conclusion, the first paragraph starts with "In this study..." Throughout the manuscript, the authors refer to it as a paper, study etc. These are informal terms and need some polishing.

 Answer: Thank you. This typo is corrected.

---

## [Decision Letter · Decision Letter 1]

24 Jan 2023

A New Cubic Transmuted Power-Function Distribution: Properties, Inference, and Applications

PONE-D-22-31175R1

Dear Dr. Afify,

We’re pleased to inform you that your manuscript has been judged scientifically suitable for publication and will be formally accepted for publication once it meets all outstanding technical requirements.

Kind regards,

Qichun Zhang, PhD

Academic Editor

PLOS ONE

Reviewers' comments:

Reviewer's Responses to Questions

**Comments to the Author**

1. If the authors have adequately addressed your comments raised in a previous round of review and you feel that this manuscript is now acceptable for publication, you may indicate that here to bypass the “Comments to the Author” section, enter your conflict of interest statement in the “Confidential to Editor” section, and submit your "Accept" recommendation.

Reviewer #1: All comments have been addressed

Reviewer #2: All comments have been addressed

2. Is the manuscript technically sound, and do the data support the conclusions?

Reviewer #1: Yes

Reviewer #2: Yes

3. Has the statistical analysis been performed appropriately and rigorously? 

Reviewer #1: Yes

Reviewer #2: Yes

4. Have the authors made all data underlying the findings in their manuscript fully available?

Reviewer #1: Yes

Reviewer #2: Yes

5. Is the manuscript presented in an intelligible fashion and written in standard English?

Reviewer #1: Yes

Reviewer #2: Yes

6. Review Comments to the Author

Reviewer #1: The paper is well revised and organised by the authors. I have noticed that the paper has been revised significantly. However, I recommend the paper for publication.

Reviewer #2: All of my suggestions were implemented within reason. This manuscript is now ready for publication in this journal.

7. PLOS authors have the option to publish the peer review history of their article (what does this mean?). If published, this will include your full peer review and any attached files.

Reviewer #1: **Yes: **Sohail Ahmad

Reviewer #2: No

---

## [Editor Report · Acceptance letter]

26 Jan 2023

PONE-D-22-31175R1 

A new cubic transmuted power-function distribution: properties, inference, and applications 

Dear Dr. Afify:

I'm pleased to inform you that your manuscript has been deemed suitable for publication in PLOS ONE. Congratulations! Your manuscript is now with our production department. 

Kind regards, 

on behalf of

Dr. Qichun Zhang 

Academic Editor

PLOS ONE